# Unsupervised Classification into Unknown number of classes

## Abstract

We propose a novel unsupervised classification method based on graph Laplacian. Unlike the widely used classification method, this architecture does not require the labels of data and the number of classes. Our key idea is to introduce a approximate linear map and a spectral clustering theory on the dimension reduced spaces into generative adversarial networks. Inspired by the human visual recognition system, the proposed framework can classify and also generate images as the human brains do. We build an approximate linear connector network $C$ analogous to the cerebral cortex, between the discriminator $D$ and the generator $G$. The connector network allows us to estimate the unknown number of classes. Estimating the number of classes is one of the challenging researches in the unsupervised learning, especially in spectral clustering. The proposed method can also classify the images by using the estimated number of classes. Therefore, we define our method as an unsupervised classification method.

## 1 Introduction

Image classification has been one of the most important research in machine learning and artificial intelligence. For the predefined number $k$ of classes, the classification models based on deep learning have performed well that the test accuracy is better than humans (Geirhos et al., 2017; He et al., 2016; Russakovsky et al., 2015; Szegedy et al., 2015). This result is, however, true only when the number of classes, $k$ is predefined and we use the supervised learning with entire ground truth labels. Humans learn visual information by memorizing the compressed information (features) in the brain, reconstruct the object, and recognize the difference between objects. This capability allows humans to classify data where the number of classes is unknown into the reasonable numbers of classes. In this paper, we propose a method to mimic such human capability.

The main contribution of the proposed framework consists of two parts. We first show that there exists a approximate linear map between two separated neural networks of Generative Adversarial Networks (GANs). Secondly, we prove that the number of classes in a latent space is the same as the number of classes in the original space in terms of the multiplicity of eigenvalue 0 of a Laplacian matrix in spectral clustering theory. By combining the results, We derive a novel unsupervised classification framework, which estimates the number of classes $k$ and then extract $k$ vectors representing each class. With the framework, the approximate linear map between two networks can induce the information of connected components in a data graph.

The latent variable model is commonly used in the generative models. Variational auto-encoders (VAEs) (Kingma & Welling, 2013) and GANs (Goodfellow et al., 2014) are a few of the most famous cases that learn the data distribution successfully. GANs randomly samples the latent variables that are used to train the data distribution so we cannot figure out which latent variable generates a result that we are interested in. To avoid the drawback by random sampling, auxiliary conditions on latent variables were introduced (Chen et al., 2016; Mirza & Osindero, 2014). Introducing a label information into the condition of the estimated data probability is highly effective in classification where a data point $x$ came from, however this method is not a unsupervised learning.

Spectral clustering is one of the most popular clustering algorithm that outperforms other well-known clustering algorithms (Ng et al., 2002; Shi & Malik, 2000). Despite high performance, spectral clustering is difficult to use because it requires the full dataset that have very large $N$ data points. In this case, the dimension of the Laplacian matrix is $N \times N$ and it takes at least $O(N^3)$. Therefore,

we need a novel dimension reduction technique to use spectral clustering in practical applications (Elhamifar & Vidal, 2009; Ji et al., 2017; Law et al., 2017; Shaham et al., 2018).

Recent image classification methods are based on supervised learning, in which a learning model processes a dataset consisting of pairs of an image and its ground-truth label. This training dataset usually consists of more than 10 K data point, some part of which need to be used for cross-validation. It also takes a lot of costs to produce a reliable training dataset. In addition, estimating the number of classes is a key aspect of unsupervised learning. There have been a few heuristic method (Zelnik-Manor & Perona, 2005; Von Luxburg, 2007). We propose a method to find the number of classes analytically.

The main concepts of spectral clustering, GANs and the self-expressiveness property will be introduced Section 2. We prove the preliminary propositions that allow us to classify data with unsupervised learning methods in Section 3. Finally, we propose a unsupervised classification algorithm and test our algorithm in Section 4.

## 2 PRELIMINARIES

We first review cycle-consistent adversarial networks and sparse subspace clustering that we will use when extracting indicator vectors. We also overview the basic probabilistic property between a classification model and a generative model, which can be used as tools to generate indicator vectors.

### 2.1 CYCLE-CONSISTENT ADVERSARIAL NETWORKS

Generative Adversarial Networks (GANs) is a generative model that a generator network learns a data distribution implicitly (Goodfellow et al., 2014). GANs is a two player minimax game with two neural networks: a discriminator $D$ and a generator $G$. The discriminator $D(\mathbf{x})$ is a probability measure that an input $x$ follows the data distribution $p_{data}(\mathbf{x})$ rather than the generator's distribution $p_g$. The generator $G$ is a mapping from the latent space $\mathbf{z} \sim p_{\mathbf{z}}$ to the data space. This minimax game can be represented as a stochastic optimization problem with value function given by

$$\min_G \max_D V(D, G) = \mathbb{E}_{x \sim p_{data}(\mathbf{x})} \left[ \log D(\mathbf{x}) \right] + \mathbb{E}_{z \sim p_z(z)} \left[ \log \left( 1 - D \left( G \left( \mathbf{z} \right) \right) \right) \right]. \tag{1}$$

The cycle-consistent adversarial networks (cycleGANs) is a variant of GANs, which learns how to translate an representation from a source domain $S$ to a target domain $T$ without a paired dataset (Zhu et al., 2017). The cycle-consistency states that two mappings $G : S \to T$ and $F : T \to S$ encode the right permutation of images in $S$ and $T$, so that, for an arbitrary data point $s \in S$, the mapping from $S$ to $T$ and back again, $F(G(s))$, should be the same as $s$. This property encourages adversarial networks $G$ and $D$ to have a connection between generated images and input data. We adopt two cycle consistency loss for the forward map and the backward map to learn a connection between two high-level representation domain in $G$ and $D$.

### 2.2 SPARSE SUBSPACE CLUSTERING

Sparse subspace clustering is a clustering algorithm that is based on spectral clustering. Spectral clustering has many variants depending on the similarity measure, Gaussian kernel similarity function (Ng et al., 2002), $k$-nearest neighbor graphs, $\epsilon$-neighborhood graph. A similarity measure defined on the similarity graph $G = (V, E)$ is a pairwise metric between two data points (nodes) in a dataset and the similarity $w_{ij}$ of two data points $x_i, x_j$ is equal to or greater than 0. The similarity $w_{ij}$ is also same as $w_{ji}$ because the above similarity graph is defined as an undirected graph. The similarity matrix $W$ is composed of the elements which correspond to the similarity measure $w_{ij}$ respectively and then $W$ is symmetric. See more details about spectral clustering in (Von Luxburg, 2007).

Sparse subspace clustering (Elhamifar & Vidal, 2009) uses a novel similarity matrix that is based on the self-expressiveness property. When we assume that the data space is a union of linear subspaces $S = \cup_{l=1}^{k} S_l$, we can express an arbitrary data point in the dataset as a linear combination of other data points in the dataset, where the coefficients of the data points in the same subspace are non-zero and others should be 0 (Elhamifar & Vidal, 2009; Ji et al., 2017). Formally, each data point $\mathbf{y}_i \in S$ can be reconstructed as $\mathbf{y}_i = \sum_{j=1}^{N} c_{ij} \mathbf{y}_j = \mathbf{Y} \mathbf{c}_i$, where $c_{ii} = 0$, N is the size of the dataset,

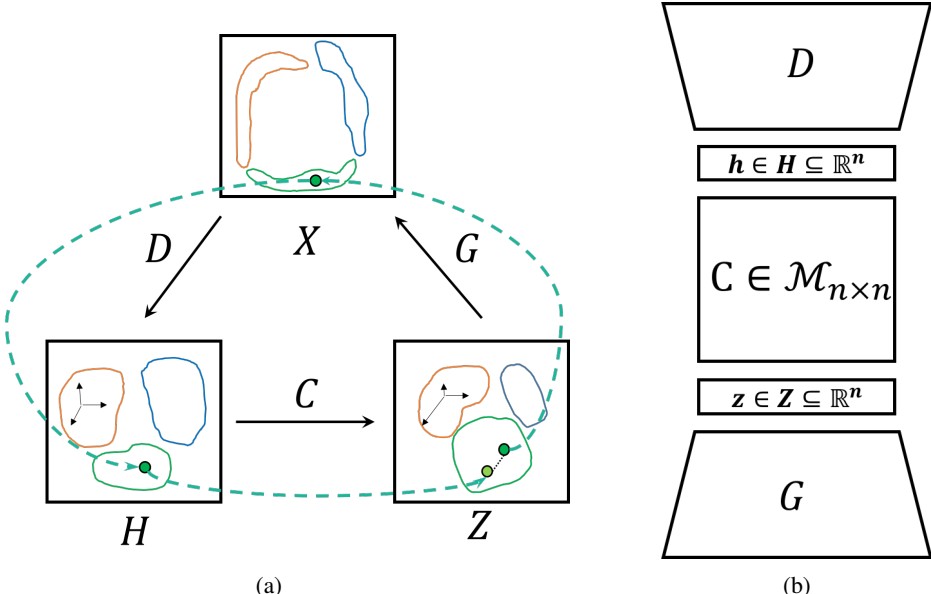

Figure 1: (a) shows the mappings relation in our proposed method. (b) shows the architecutre of the our networks.

$\mathbf{c}_i := [c_{i1} \; c_{i2} \; \cdots \; c_{iN}]^T$ is the coefficient vector of $\mathbf{y}_i$ and $\mathbf{Y} = [\mathbf{y}_1 \; \mathbf{y}_2 \; \cdots \; \mathbf{y}_N]$ is the dictionary matrix. The sparse solution of $\mathbf{c}_i$ indicates that the node $i$ is in the same subspace as the node $j$ if the element $c_{ij}$ is non-zero. In this paper, we expand the notion of the self-expressiveness property to reduce the dimensionality of $L$ in Section 3.3.

## 3 GRAPH LAPLACIAN OF THE CONNECTOR NETWORK

We first introduce our proposed architecture of adversarial networks. As an autoencoder point of view, we can consider two adversarial networks as autoencoder. We build a connector network between two networks to improve cycle consistency, and extract the information of connected components (as shown in Figure 1-b). By training with the cycle-consistency loss, we can make the tuples $(Z, X, H)$ for each data point $X$ in a given dataset. Then, we show the relation between the spectral clustering and unsupervised classification. In order to show the relationship, we prove the dimension reduction of the mutual expressiveness property in Section 3.3.

### 3.1 DISENTANGLING UNFOLDS AND EXPANDING THE VOLUME OF MANIFOLD

Deep neural networks learn a high-level abstract representation of a data manifold and a well trained abstraction can disentangle the underlying factors of variation (Bengio et al., 2007). These properties allow interpolation between data samples more naturally and smoothly. (Bengio et al., 2013) studied the disentangling effect and proposed three hypotheses on the shape of manifolds in different representation domains. These hypotheses are verified by several empirical results (Goodfellow et al., 2009; Glorot et al., 2011). The three hypotheses are summerized as follows.

- The deeper representations can better disentangle the underlying factors of variation.
- Disentangled representations unfold the manifolds near which raw data concentrates irregularly, and expand the relative volume occupied by high-probability points near these manifolds. Then these representations are with greater convexity.
- The underlying class factors of variations will be better disentangled than other factors, so the deeper layer has better discriminant capability of classes.

In our proposed architecture, there are two high-level (deeper) representation spaces which are called as feature spaces. These feature spaces fill the space more uniformly than the pixel space $X$. $H$ and

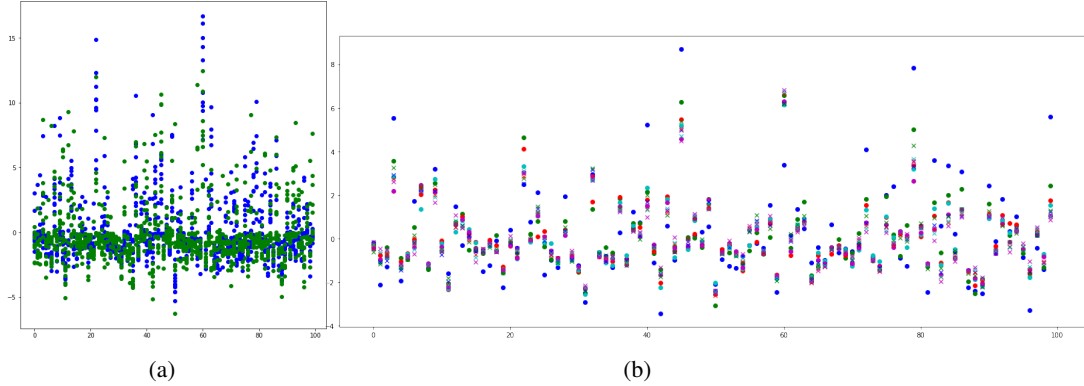

Figure 2: (a) shows the density of h for 1 and 2 MNIST, (b) shows the mean of $h$ for entire MNIST classes

$Z$ also have smoother density. Therefore, $H$ and $Z$ have greater convexity than $X$ and it permits linear combination on these two manifolds in the feature spaces $H$ and $Z$, are the two manifolds are more convex than the manifold in the pixel space $X$. By manifold hypothesis, different class manifolds are well-separated by regions of very low density (Cayton, 2005). The feature spaces are high-level representation spaces so that the density is unfolded and a linear combination of features generates amore natural result in the pixel space Bengio (2014).

Now, we can define the effective basis of the subspace for each class. In the feature spaces, each class manifold is assumed to have the enough convexity. The class manifolds are well-separated, and show good discriminant capability. Using both properties, we can build the effective basis for each class subspaces and generate the original space by a direct sum of these subspaces, where the subspaces are orthogonal. We empirically tested an effective basis as shown in figure 3. Randomly sampled features in a subspace of the data space can be an effective basis, and are possible to generate all data points in the same subspace. The features in a different subspace cannot generate the given data samples. We can estimate the effective dimension with the number of the effective basis. We then can generate orthogonal subspaces of the entire classes if the effective dimension is small enough than the dimension of feature spaces.

## 3.2 CONNECTOR NETWORK AND CYCLE CONSISTENCY

We will study the relationship among 3 different spaces, $Z$, $X$ and $H$ in the proposed architecture. As shown in figure 1-(b), we can consider two adversarial networks as a decoder and an encoder of an auto-encoder respectively (Berthelot et al., 2017; Zhao et al., 2016). The discriminator network $D$ encodes the pixel space $X$ to the feature space $H$, and the generator network $G$ decodes the feature space $Z$ to $X$. We define the connector network $C$ which maps from $H$ to $Z$. The map $C$ between two feature spaces will be trained by the cycle consistency loss to obtain the tuples $(Z, X, H)$ with the correct permutation, where all the elements are in the same-class manifold and shares the same learned features. Cycle consistency loss tends to minimize the difference between an arbitrary point and the returned point after traveling other representation domains. This cycle-consistency objective permits inverse maps between all representation spaces. When we apply the forward cycle-consistency loss $\|z - C(D(G(z)))\|_1$ and the backward cycle consistency loss $\|x - G(C(D(x)))\|_1$, the maps between all pairs of neighboring spaces become unique. Therefore, the feature-invariant one-to-one maps among $G, C, D$ can be induced. All elements of an arbitrary tuple $(Z, X, H)$ share same class features as shown In figure 1-(a). We now have the following assumption.

**Assumption 3.1.** *All elements of the cycle consistent tuple $(Z, X, H)$ shares the identical factors, such as class factor and other factors (e.g. rotation and brightness).*

As shown in fig 3-(c, d), we can conjecture that each class subset is closed and the linear combinations in the certain class subset are also in the class because each class manifold is unfolded and expanded in $Z$ and $H$. In addition, we showed there exists the effective basis for each class. By

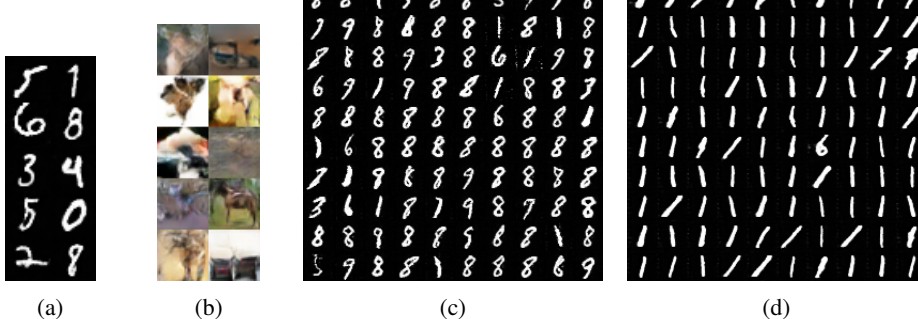

(a)         (b)             (c)              (d)

Figure 3: (a), (b) the generate images using the indicator vectors of Laplacian on $C$ for MNIST and CIFAR10, (c) describes the linear combination using class 2 to generate 1 but failed. (d) is the generated image by the linear combination of the right features $h$ for 1

the hypotheses (Bengio et al., 2013), we know that each class manifold has the compactness and the convexity. Now we can assume the following assumption.

**Assumption 3.2.** *Each class subspace $H_i$, where $H_i \subset H$ are orthogonal, so as $Z_i$ where $Z_i \subset Z$. We define sets for each class $S_{H_i}$ and $S_{Z_i}$ in $H_i, Z_i$ respectively to be the the class manifolds which lies on the corresponding class subspaces. The sets $S_{H_i}$ and $S_{Z_i}$ are compact and convex.*

This assumption is more tight and realistic than the union of linear subspaces in Elhamifar & Vidal (2009). We can also conjecture that the feature space $H$ and $Z$ are descried as the direct sum of $H_i$ and $Z_i$ respectively. Now, we can show the existence of an approximate linear map $C$ from $H$ to $Z$.

**Lemma 3.1.** *There exists a compact and convex linear map $C_i$ from $S_{H_i}$ to $S_{Z_i}$, If $\|C_i\|$ is closed.*

*Proof.*

1. By the assumption 3.2, the class subset $S_{H_i}$ and $S_{Z_i}$ are convex. Let $h_1, h_2 \in S_{H_i}$ and $z_1 = C_i h_1, z_2 = C_i h_2$. We know that $C_i$ is linear, then

$$tz_1 + (1-t)z_2 = tC_i h_1 + (1-t)C_i h_2 = C_i(th_1 + (1-t)h_2). \tag{2}$$

   $th_1 + (1-t)h_2$ is also in $S_{H_i}$ for $t \in [0,1]$. Also, $tz_1 + (1-t)z_2$ is in $S_{Z_i}$ because $S_{Z_i}$ is convex. Therefore $C_i$ is a convex linear map for all $i$.

2. If $S_{H_i}$ is compact, then for $h \in S_{H_i}$, $\|h\|$ is closed. Then $z \in S_{Z_i}$ is closed, because

$$\|z\| = \|C_i h\| \leq \|C_i\|\|h\|. \tag{3}$$

   We know that $C_i$ and $h$ are finite. Therefore $S_{Z_i}$ is also compact.

**Theorem 3.1.** *By the assumption that the data space are the direct sum of orthogonal class subspaces, the map from H to Z can be written as a block matrix, of which block component consists of $C_i$. C can maps from all class subset $S_{H_i}$ to $S_{Z_i}$.*

*Proof.* By Lemma. 3.1., $C_i$ is a compact and convex linear map from $H_i$ to $Z_i$. $H$ and $Z$ are the direct sum of the orthogonal subspaces $H_i$ and $Z_i$ respectively. Therefore $C$ can be represented as a block diagonal matrix for some ordered basis.

## 3.3 MUTUAL EXPRESSIVENESS PROPERTY AND DIMENSION REDUCTION

As we discussed in Section 2, the self-expressiveness property shows that we can express an arbitrary data point as a linear combination of other $N$ data points. By modifying this property slightly, we can write each data point $x$ as a linear combination of $N$ sampled data points $\alpha = \{\mathbf{x}_1, \mathbf{x}_2, \cdots, \mathbf{x}_N\}$ from the data space $X$. This fact allows us to generate a set of feature variables, which satisfies the self-expressiveness property. The self-expressiveness property implies that the generated data points are in the class manifolds. Now, we relax the self-expressiveness property to generalize the problem. We divide the latent space $H$ into $H_1, \cdots, H_l$ where the self-expressiveness only satisfies in each

$H_i$ not in $H - H_i$. We also know that it is possible to discard redundant coefficients in the self-expressiveness expression by optimizing the matrix of coefficients (Elhamifar & Vidal, 2009), so there exists at least one sparse solution of $x$ where most coefficients are zero. By observing this property, it is expected that the number of non zero coefficients approaches the dimension of latent space, $m$. That is because $N$ is much higher than $\dim X$, and $X$ is a $m$-dimensional manifold. Image clustering methods, for instance, treat a dataset where the number of data points is much larger than the dimension of data points so we can optimize a solution to have sufficiently many zero coefficients in the self-expressive linear combination.

**Theorem 3.2.** *From now on, $Z_i' = C_i' H_i'$ where $Z_i' = (z_1, \cdots, z_{m_i})^T$ and $H_i' = (h_1, \cdots, h_{m_i})^T$.*

*Proof.* By an observation on GANs, we have $N_i$ feature variables $\beta_i = \{h_{ij}\}_{1 \leq j \leq N_i}$ for each $1 \leq i \leq l$ where $D(x_i) = h_i$ and $N = N_1 + \cdots + N_l$. By the self-expressiveness property in $H_i$, we can express each $h_i \in H_i$ by the linear combination of $\beta_i' = \{h_1, h_2 \cdots, h_{m_i}, h_{m_i+1}, \cdots h_{N_i}\}$ where $\beta_i'$ is a reordered set of $\beta_i$ and the coefficents of the linear combination $\{c_{m_i+1}, c_{m_i+2}, \cdots, c_{Ni}\}$ are zero. In other words, the ordered set $\gamma_i = \{h_1, \cdots, h_{m_i}\}$ is enough to represent all points in the class manifold $X_i$ and $Z_i$, which correspond to $(Z_i, X_i, H_i)$, i.e. $span(\gamma_i) = H_i$. Hence, we can choose $C_i'$ by an $m_i \times m_i$ real matrix. $\square$

This theorem is empirically shown by the our effective basis conjecture. Now, we can conclude that the $\dim Z_i = \dim H_i = m_i \ll N_i$. We define the connector network $C$ as a $(\sum m_i) \times (\sum m_i)$ matrix. The similarity matrix is denoted by $W = |C| + |C^T|$.

**Theorem 3.3.** *Let $G$ be an undirected graph with non-negative weights. Then, the number of connected components $A_1, \cdots, A_k$ in $G$ is the same as the multiplicity $k$ of the eigenvalue 0 of $L$. If $W$ is a $k$ block diagonal matrix, then the multiplicity of the eigenvalue 0 of its Laplacian is also $k$.*

*Proof.* the proof in (Von Luxburg, 2007). By the definition of $W$, the number of sub-block matrices in $W$ is the same as the number of connected components. $\square$

We conclude that the numbers of classes are identical between spectral clustering and our method.

## 3.4 UNSUPERVISED CLASSIFICATION

Now, we consider a unsupervised classification problem where there is no ground-truth label. For example, when the cross-entropy loss is applied in the supervised classification, we need pairs of input data and ground-truth probability mass $p = [0, , 1, \cdots, 0]$, which contains a single 1 at the $i$th element, i.e. the class index $c = i$, to estimate the difference between the prediction and the ground-truth. We can formally write the probabilistic interpretation as

$$P(c = i|\mathbf{x}) = \frac{P(c = i)P(\mathbf{x}|c = i)}{P(\mathbf{x})} \tag{4}$$

, where $\mathbf{x}$ is a data point. If we assume that the numbers of data points in the whole classes are same, then $P(c)$ is uniform. Then, for the given point $\mathbf{x}$, $P(\mathbf{x})$ is fixed so the we can measure $P(c|\mathbf{x})$ by the estimating the likelihood $P(\mathbf{x}|c)$. In Section 3.3, we showed that the indicator vectors of the connected components are the eigenvectors of the Laplacian matrix $L$, which are corresponding to the eigenvalue 0. The components of the indicator vectors span the eigenspaces of the matched connected components respectively. The orders of components for the Laplacian matrix $L_C$ and the connector network $C$ are identical by the definition in Section 3.3. Therefore, for all connected component $A_i$, the indicator vectors $\mathbf{1}_{A_i} = h \in H_i$ spans the class subspace $H_i$. The indicator vector itself, however, might not be in the class manifold because it can exist outside the boundary of the manifold. We find out that the changes of norms between $H$ and $Z$ are different by the class. The connector network can be considered as a singular value decomposition for each class. This implies that each class data points move differently by the class. We define the mean vector of a class in $H$ as $h'$ and the Gaussian noise $n$.

$$\|z\|^2 = \|Ch\|^2 = h^T C^T C h \tag{5}$$

$$= (h' + n)^T C^T C (h' + n) \tag{6}$$

The expectation of $\|z\|^2$ is,

$$E[\|z\|^2] = h'^T C^T C h' + E[n^T C^T C n] \tag{7}$$

$$= h'^T C^T C h' + E[n_Z^T n_Z] \tag{8}$$

, where $n_Z$ is the scaled noise in $Z$. We now assume that the distribution of $\|z\|^2$ follows a Gaussian with a variance $\sigma_{n_Z}$. Then, we can calculate the probability that $\|z\|^2$ is in the certain class. We also know that $B = C^T C$ is a normal operator, then $B$ satisfies the following theorem.

**Theorem 3.4.** $|x^T B x| \leq \max\{|\lambda| : \lambda \text{ is an eigenvalue of } B\}\|x\|_2^2$

*Proof.* See in Chapter 5.6 of (Horn et al., 1990).

By ordering the eigenvalue of $B$, we can find out the interval that $\|z\|^2$ lies in. This implies that the maximum eigenvalue of the sub-block matrix in $B$ for $z$ is in the above interval. This eigenvalue works as the indicator of classifying class. Therefore we can build of Gaussian distributions for $\|z\|^2$ with the mean $\lambda\|h\|^2$ and the standard deviation $\sigma_{n_Z}$.

# 4 UNSUPERVISED CLASSIFICATION INTO UNKNOWN $k$ CLASSES

In this section, we describe the loss function and the training algorithm of unsupervised classification into unknown $k$ classes. It is known that estimating the number of classes $k$ is hard and heuristic (Von Luxburg, 2007). We find the empirical observation on the hint of $k$. Using the observation, we suggest a conjecture that helps to estimate the number of classes $k$ in a certain case.

---

**Algorithm 1** Unsupervised Classification into Unknown $k$ Classes (UCUC)

---

1: **for** the number of iterations **do**
2:   Sample minibatch of $m$ latent vectors $\mathbf{z}^{(0)}, \cdots, \mathbf{z}^{(m)}$ from prior $p_g(z)$
3:   Sample minibatch of $m$ data points $\mathbf{x}^{(0)}, \cdots, \mathbf{x}^{(m)}$ from dataset
4:   Update the discriminator by ascending :

$$\nabla_{\theta_d} \frac{1}{m} \sum_{i=1}^{m} \left[ \log D\left(\mathbf{x}^{(i)}\right) + \log\left(1 - D\left(G\left(\mathbf{z}^{(i)}\right)\right)\right) \right] + \lambda_D \left\|\mathbf{x}^{(i)} - G\left(C\left(D\left(\mathbf{x}^{(i)}\right)\right)\right)\right\|_1 \tag{9}$$

5:   Update the generator and connect by descending their stochastic gradients:

$$\nabla_{\theta_g} \frac{1}{m} \sum_{i=1}^{m} \left[ \log\left(1 - D\left(G\left(\mathbf{z}^{(i)}\right)\right)\right) \right] + \lambda_G \left\|\mathbf{z}^{(i)} - C\left(D\left(G\left(\mathbf{z}^{(i)}\right)\right)\right)\right\|_1 \tag{10}$$

6:   Update the connector network $C$ with two cycle-consistency loss

$$\nabla_C \left[ \left\|\mathbf{z}^{(i)} - C\left(D\left(G\left(\mathbf{z}^{(i)}\right)\right)\right)\right\|_1 + \left\|\mathbf{x}^{(i)} - G\left(C\left(D\left(\mathbf{x}^{(i)}\right)\right)\right)\right\|_1 \right] \tag{11}$$

7: **end for**
8: Compute the moving average of each $i$th eigenvalue of normalized symmetric Laplacian matrix $L_{sym} = D^{-\frac{1}{2}} L D^{-\frac{1}{2}}$ and find the estimated number $\hat{k}$ by computing the second local minimum

9: Calculate the eigenvalues of $C^T C$ and find a matched class for an input

---

## 4.1 ESTIMATING THE NUMBER OF CLASSES $k$

We found an interesting observation on the number of classes $k$. The eigenvalue $\lambda$ of the graph Laplacian matrix equals to Ncut of the given graph (Shi & Malik, 2000). In the ideal cases, we can assume that $L$ is a block-diagonal matrix and each of blocks $L_i$ is a proper sub-graph Laplacian. If there is no connection between subgraphs, then $L$ is a perfect block-diagonal matrix and the multiplicity of the eigenvalue 0 is the same as the number of blocks. In the real world dataset, there exists quite large off-diagonal noise in the similarity matrix, as shown in the previous result. These non-zero off-diagonal elements make the eigenvalue zero only at the first eigenvalue and increase the Ncut of the graph gradually. We can consider the $i$th eigenvalue as the Ncut by the block diagonal matrix with $i$ blocks, which is applied on the perturbed block diagonal matrix of which the underlying connected component is $k$. We now have the following conjecture on the pattern of eigenvalues. We assume that the off-diagonal elements have the same value and the dimensions of

all eigenspaces are the same. Now, we estimate the number of classes $k$ by starting from $m \ll k$. We can consider the difference of Ncut between $k$-block diagonal matrix and $m$-block diagonal matrix as the the area of $S_m - S_m \cap S_k$, where $S_m = \frac{n^2}{m}$ and $S_k = \frac{n^2}{k}$ when the matrix size is $n$. As $m$ increases, the difference of Ncut increases, but at $k = m$ the value have the very small local minimum becuase two matrices are perfectly overlapped. Using this property we can derive the function of the difference of the Ncut over m as

$$S_m \cap S_k = n^2 \left( \frac{1}{k^2}(k - m + 1) + \sum_{i=1}^{m-1} \left[ \left( \frac{i}{m} - \lfloor \frac{ik}{m} \rfloor \frac{1}{k} \right)^2 + \left( (1 + \lfloor \frac{ik}{m} \rfloor) \frac{1}{k} - \frac{i}{m} \right)^2 \right], \quad (12)$$

where $m \leq k$. if $m > k$, and switch $m$ and $k$. This overlapped area is $\frac{n^2}{k}$ at $m = k$ trivially. The figure of the theoretical conjecture is in Figure-4a. This result can predict the empirical result of the pattern of the eigenvalue. Therefore, we can conjecture that the underlying number of classes is at the second deep local minimum.

## 5 EVALUATION

We introduce the setting of experiments and the architecture of our adversarial networks (Radford et al., 2015). For the smaller size of dataset, such as MNIST and CIFAR10, we reduce the size of kernels properly. We have the result the expectation of the estimated number of k is 10.2 for MNIST and 10.1 for CIFAR10. Now we use the unsupervised clustering accuracy (ACC) (Cai et al., 2011; Shaham et al., 2018).

$$ACC(l, c) = \frac{1}{n} \max_{\pi \in \Pi} \sum_{i=1}^{n} 1\{l_i = \pi(c_i)\}, \quad (13)$$

where $l_i$ is true label, $c_i$ is the ground-truth label, and $\pi$ is the collection of permutations.

| Algorithm | ACC (MNIST) |
|---|---|
| spectral clustering | .717 |
| SpectralNet | .971 |
| IMSAT | .984 |
| UCUC(ours) | .735 |

Table 1: Other methods have the ground-truth number of classes. UCUC is not given the number of classes. Other results are reported in (Shaham et al., 2018).

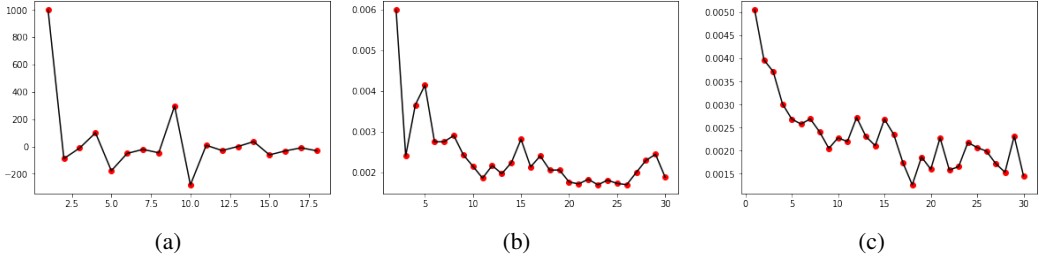

(a)         (b)         (c)

Figure 4: a shows theoretical result of the estimated eigengap using the overlapped area of block-diagonal matrices and Figure (b) and (c) shows the eigengap $|\lambda_k - \lambda_{k-1}|$ of the MNIST and CIFAR10.

## 6 CONCLUSION

We have proposed an unsupervised classification method that can automatically estimate the unknown number of classes $k$. Our method provides the learned class-features. The proposed method saves the tremendous cost of producing ground-truth labels in a large dataset.

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

# A    SUPPLEMENTARY MATERIALS

## A.A    MODEL ARCHITECTURE

| DCGAN | Generator | Discriminator |
|---|---|---|
| MNIST | fc 100
fc 1024
fc 2048
conv [5,5,256]
relu, batch_norm
conv [4,4,128]
relu, batch_norm
conv [5,5,64]
relu, batch_norm
conv [5,5,1]
tanh, batch_norm | conv [5,5,32]
lrelu, batch_norm
conv [5,5,64]
lrelu, batch_norm
conv [5,5,128]
lrelu, batch_norm
conv [5,5,256]
lrelu, batch_norm
fc 100 (H)
lrelu
fc 1
sigmoid |
| CIFAR10 | fc 100
fc 2048
fc 4096
conv [5,5,512]
relu, batch_norm
conv [5,5,256]
relu, batch_norm
conv [5,5,128]
relu, batch_norm
conv [5,5,3]
tanh, batch_norm | conv [5,5,32]
lrelu, batch_norm
conv [5,5,64]
lrelu, batch_norm
conv [5,5,128]
lrelu, batch_norm
conv [5,5,256]
lrelu, batch_norm
fc 100 (H)
lrelu
fc 1
sigmoid |

Table 2: Model Architecture in the MNIST and CIFAR10 experiments.

| | |
|---|---|
| batch size | 64 |
| $\dim z$ | 100 |
| learning rate, $\beta_1$ | 0.0002, 0.5 |
| $\lambda_G, \lambda_D$ | 0.5, 0.5 |

Table 3: The hyperparameters for the networks

## A.B    ADDITIONAL EXPERIMENTAL RESULTS

In the latent space $H, Z$, the each class subset has a Gaussian distribution, which implies the subset is convex. We observed that the connector networks $C_i$ are not identical among the whole classes. Each connector network $C_i$ is decomposed into an unique singular value decomposition matrices. Therefore, we can extract the difference for each class by analyzing the eigenvalues in Section 3.4.

We can observe that $W$ is a near block diagonal matrix, of which off-diagonal elements are not zero. This leads to the summation of some residue weights which should be zeros theoretically. When the number of connected components are matched, those residues are no longer summed up. Therefore, the deep local minimum in the eigengap exists.

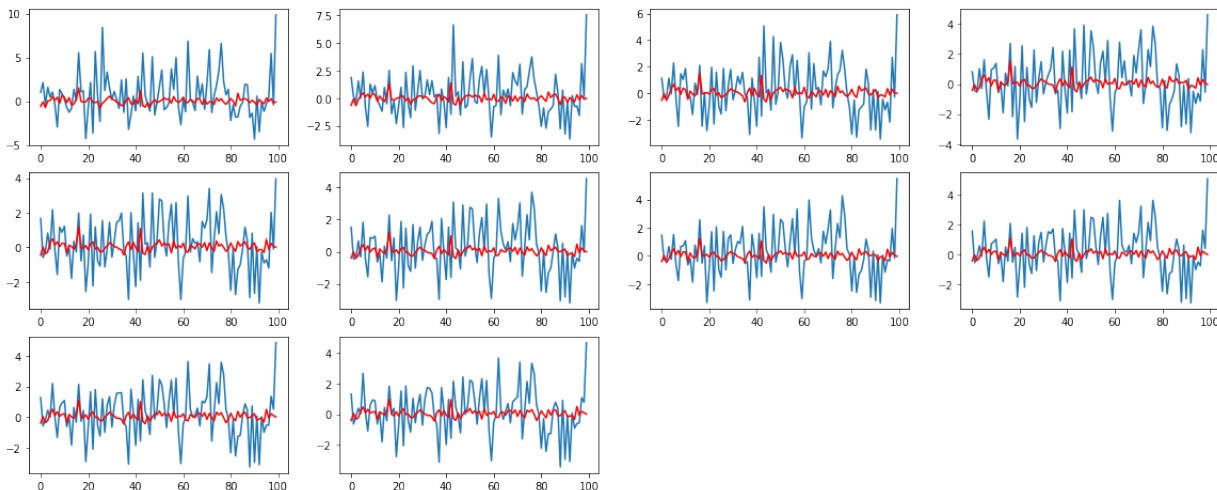

Figure 5: In the some range of noise, we can always generate the same class image. The magnitudes of noise for each class are different.

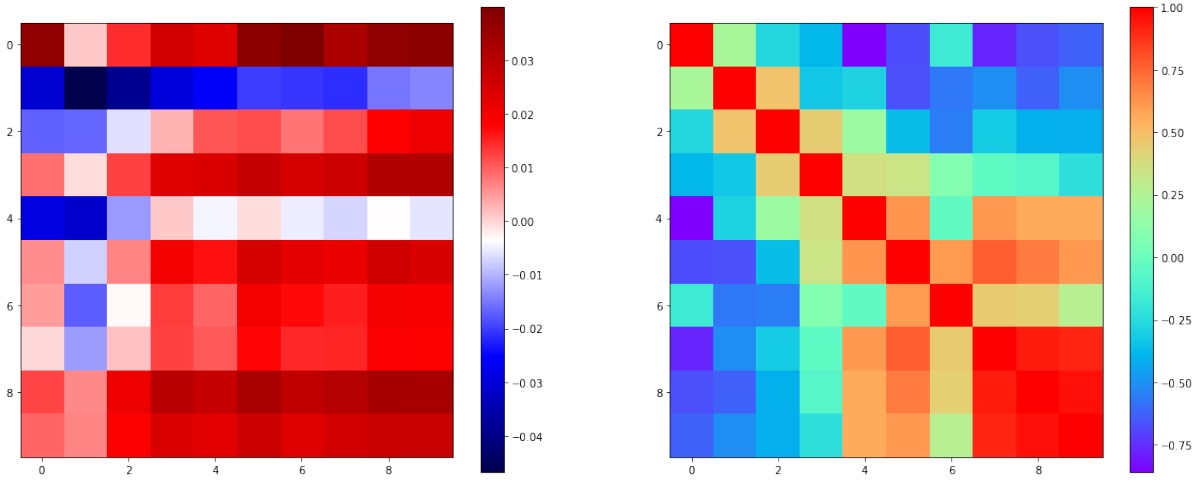

Figure 6: We sample $Z$ from the normal distribution, so the norm of $z$ is smaller than $h$. In additino, we can observe that $C$ changes not only the magnitude of the vector but also the angle.

Figure 7: The left figure is the cosine values between the mean vector of each classes $\bar{h}_i, \bar{z}_j$. The right figure is the cosine values between $\bar{h}_i, \bar{h}_j$.

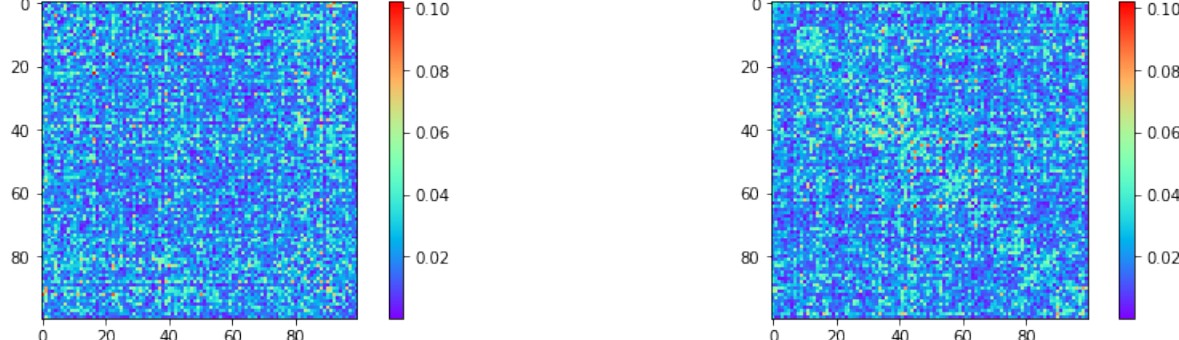

Figure 8: The left figure is the initial similarity matrix $W$, The right figure is the right permutated similarity matrix.

