# OpenReview forum: "Unsupervised classification into unknown number of classes"
_ICLR.cc/2019/Conference_

### Official Review · AnonReviewer3 · 2018-11-03
**lack of sufficient experiments**

**Rating:** 5
**Confidence:** 4

**Review:**

The manuscript proposes a method for unsupervised learning with unknown class number k. The problem is classical and important. The proposed method is interesting and novel, but the experiments are not convincing. In detail, it did not compare other methods in the experiments.
Pros: clear description and novelty of the method
Cons: insufficient experiments.

---

> ### Author Response · Authors · 2018-12-01
> **Add the additional experiment results**
>
> We appreciate your feedback. We first notice that we slightly change little details to classify images.
>
> 1. the experiments are not convincing
> We wrote the classification part ambiguously. It was our fault. In the revised version, we describe how to classify images into the estimated number of classes. We use the distinct eigenvalues of C^TC for each class in section 3.4 to show changes of the magnitude of latent vectors among classes are different. We also write the criteria of measuring the performance of unsupervised clustering, ACC, which we use in the experiment. We are now doing more experiments with a new version of classification methods as I mentioned. We expect that we can attach more results in the camera-ready if we are accepted. We also add supplementary materials in appendix. which helps to understand how our method works.
>
> 2. it did not compare other methods in the experiments.
> We add three other methods. other methods do clustering with the fixed and known number of classes. So, it might be the reason why two of them shows the better performance than ours.

---

### Official Review · AnonReviewer1 · 2018-11-06
**Maybe an interesting paper but many parts are unclear**

**Rating:** 4
**Confidence:** 4

**Review:**

This paper develops an unsupervised classification algorithm using the idea of CycleGAN. Specifically, it constructs a piece-wise linear mapping (the Connector Network) between the discriminator network and the generator network. The learning objective is based on the cycle-consistency loss. Experiments show that it can achieve reasonable loss. This paper addresses an important problem, namely, unsupervised image classification, and may present interesting ideas. However, the paper is not in a good shape for publication in its current form.

First, the paper is not well written and many of the key ideas are not clear. It devotes more than half of the pages to the review of the preliminary materials in Sections 2-3 while only briefly explained the main algorithm in Section 4. Many of the details are missing. For example, why L1-loss is used in (5)-(7) in Algorithm 1? What is the “random-walk Laplacian matrix L_{sym}” (never defined)? More importantly, it seems that Section 3.4 is a key section to explain how to perform unsupervised classification. However, the ideas (regarding the footprints and footprint mask etc.) are totally unclear. It is assumed that all different classes have equal probabilities. In this setting, it is unclear (in its intuition) why it is possible to assign a cluster index to its true class labels. What is the key signal in the data that enables the algorithm to relate different clusters to their corresponding classes, especially when all classes have equal probability? Furthermore, it is not clear why the mapping from H to Z can be written as a sum of C_1,…,C_k in Proposition 3.2. If the final mapping is piece-wise linear, how can it be written as a sum of linear mappings? Similar question arises in the first paragraph of Section 4.1: if the connector network is constructed as a piecewise linear function (as stated earlier in the paper in abstract and introduction), then how can it be written as a matrix? (Only linear mapping can be expressed as a matrix.)

Second, the experiment is insufficient. None of the experiment details are presented in the paper. Only the final accuracy of 0.874 is given without any further details. What is the model architecture and size? More experimental analysis should be presented. For example, there are many different hyperparameters in the algorithms. How are the \lambda_D, \lambda_G chosen when there is no labeled validation set? How sensitive is the algorithm to different model architecture and model size? Furthermore, none of the baselines results are presented and compared against.

A lot of related works are missing. There have been a lot of emerging works related to unsupervised classification recently, which should be discussed and compared:
[1] G. Lample, L. Denoyer, and M. Ranzato.  Unsupervised machine translation using monolingual corpora only. ICLR, 2018.
[2] M. Artetxe, G. Labaka, E. Agirre, and K. Cho.  Unsupervised neural machine translation. ICLR, 2018.
[3] Y. Liu, J. Chen, and L. Deng.  Unsupervised sequence classification using sequential output statistics. NIPS, 2017
[4] A. Gupta, A. Vedaldi, A. Zisserman. Learning to Read by Spelling: Towards Unsupervised Text Recognition. arXiv:1809.08675, 2018.
[5] G. Lample, M. Ott, A. Conneau, L. Denoyer, M. Ranzato. Phrase-based & neural unsupervised machine translation. EMNLP 2018.

The presentation of the paper should be significantly improved as it is currently hard to read due to many grammar and English usage issues as well as other unclear statements. Just to name a few examples below:
-	(1st paragraph of Introduction): “…imagine the learned objectconstruct…”
-	The last paragraph in Section 1 is not in the right position and should be placed somewhere else in the introduction.
-	In the first paragraph of Section 2.2, “one of the clustering algorithm” should be “one of the clustering algorithms”.
-	In the first paragraph of Section 3, it is not clear what it means by “we can make the tuples (Z,X,H) for the whole dataset”.
-	At the end of the first paragraph of Section 3, there is a missing reference in “network in section()”.
-	In the third paragraph on page 4, there is a grammar issue in “H and Z have greater than convexity than X…” and in “it allows the linear combination on these two manifolds in the feature spaces H and Z are and”.
-	In the first paragraph of Section 3.2, it is not clear what it means by “two feature spaces will be trained by the cycle consistency loss to obtain the tuples (Z,X,H) with the correct permutation, where all of elements is in the same-class manifold and shares same learned features.”
-	In the first paragraph of page 5, “cycle-consistency loss z C(D(G(z))) and backward cycle consistency loss x G(C(D(x)))” does not read well. It sounds like z is the loss C(D(G(z))) and x is the loss G(C(D(x)))?
-	Typo in Figure 4: “a shows” should be “(a) shows”.

---

> ### Author Response · Authors · 2018-12-01
> **first comment**
>
> Thanks for giving us your feedback. As you mentioned in your review, we did not write enough explanations We revised our paper based on your feedback. First, we notice that we slightly change the method of details of classifying method.
>
>
> 1. problems about writing
>  The list below is the errors of my paper in your feedback. We corrected all errors and rebalance the amount of section 2 and section 3.
>
> - not good shape and grammar issue
> - the balance of amount between section 2 and section 3
> - typo and errors
>
> 2. related works
> You refer the very good papers which are related to our work. However, by the 8 pages limit, we did not attach in the revised version. Because these work does not cover the unsupervised classification for images. Nevertheless, we are going to deal with this work in the camera-ready version for related works of unsupervised learning if we are accepted.
>
> 3. why L1-loss is used in (5)-(7) in Algorithm 1?
> This is because the cycle consistency, which is used in cycleGAN.  In addition,  L2 and L_inf loss cannot get us a sparse solution, which is the key to divide C into sub block matrices.
>
> 4. piecewise linear map
> Thanks for your feedback, we change piecewise linear map to approximate linear map. When we first start the research, we think that we can express C as the piecewise linear map and we can approximate this as a linear map. However, we remove that part, and introduce the assumption 3.2 lemma 3.1 to make our method more clear.
>
> 5. What is the key signal in the data that enables the algorithm to relate different clusters to their corresponding classes, especially when all classes have equal probability?
> The classification datasets usually have the same numbers of datapoints for each class. So we can assume that the probability that a datapoint in a class is equal. This means that we can use the likelihood itself as the probability that a given datapoint belong to a class.

---

> ### Author Response · Authors · 2018-12-01
> **second comment**
>
> 6. why the mapping from H to Z can be written as a sum of C_1,…,C_k in Proposition 3.2.
> First I will tell the existence of linear map C, and show why C has C_1, …, C_k as the diagonal sub-blocks.
>
> To show the existence of a convex-compact linear map between two class manifolds in H and in Z, we introduce Assumption 3.2. Assumption 3.2 is more realistic and tight than the assumption that the data space is the union of linear subspaces. Each class manifold in the high abstraction space (H and Z) has convexity and compactness as discussed in Bengio et al. 2013. In addition, we showed that class manifolds are disjoint, so we cannot generate other class samples by using a few samples in a class. It implies that the subspaces that the two different class manifolds(exactly they are subsets) lie on are orthogonal, and sums of the dimension of subspaces H_i and Z_i are dim(H) and dim(Z) respectively. Therefore, we can conclude that H and Z are the direct sums of H_i and Z_i.
>
> By introducing the lemma 3.1. we know that C_i is convex and compact linear map. Then, we can prove the existence of a convex-compact linear map. This linear map guarantees at least between S_H_i and S_Z_i. Even if C_i does not hold compactness and convexity outside the the class subset, it does not matter. Because the outside of class manifolds is nothing.
>
> Now, we can find a linear map from H to Z, which satisfies in the union of class manifolds, by using Theorem 3.2. In addition, we now know that C consists of sub-block matrices C_1, …, C_k by theorem 3.1
>
>
>
> 7.. the meaning of section 3.4
> After the first submission, we find out the result by footprint mask is somewhat unstable, and the theoretical meaning is lack as you mentioned. We find another classification criterion which shares the same theoretical background.
>
> we want to find a disentangled property among each class. We can interpret each C_i for class i into a singular value decomposition. In the experiment, we observed that the changes of magnitude and angle by the linear map C are different with the class factor in appendix. This implies C_i has different distinct singular values and semi axis. Then, each block component C_i in C has different eigenvalues. Therefore, we can assign eigenvalues into block sub-matrices. These eigenvalues will be the metric for classification.
>
> The better aspect of classification than clustering is that the classification provides the score for each class that measures how much likely a given data point belongs to the class. Our method uses the eigenvalue as the classification metric. We define the Gaussian distributions of which mean is the eigenvalue of C^TC for each class. We find eigenvalues as much as the estimated number of classes by our algorithm. We calculate the probability for each class for the change of magnitude of a given data point. Then, we can find which class is most probable.

---

### Official Review · AnonReviewer4 · 2018-11-12

**Rating:** 4
**Confidence:** 2

**Review:**

------------------------------------------
Summary
------------------------------------------
This paper performs unsupervised classification where the number of classes is unknown. The main idea is to use the CycleGAN framework such that one can reconstruct the original image by first moving to latent space that represents another class (via the connector network), then moving back to the original latent space and going back into image space using a generator. Experiments are conducted on MNIST and CIFAR.

------------------------------------------
Evaluation
------------------------------------------
The paper tackles an important problem: namely, unsupervised classification (i.e. clustering). I think the use cycle-consistency loss in an auxiliary latent space is quite clever. However, experimental results are lacking. Unsupervised clustering (even when number of classes is not known) is a very well studied problem in machine learning. The authors should compare against at least a few reasonable baselines.

------------------------------------------
Presentation
------------------------------------------
I found the presentation to be somewhat wanting. Section 3 is extremely confusing and in my opinion, not well-motivated. For example, why is self-expressivity important? Why can we assume propositions 3.1 and 3.2?

---

> ### Author Response · Authors · 2018-12-01
> **first comment**
>
> Thanks for giving us your feedback. As you mentioned in your review, we did not write enough explanations about section 3. We will deal with the lack of explanation of section 3 in the end. First, we notice that we slightly change the method of details of classifying method.
>
> 1. experimental results are lacking, compare against at least a few reasonable baselines
> In the revised version, we introduced the criteria which we used. ACC measures the performance of the clustering, which is analogous to the prediction accuracy in the classification. We add some results of other methods. We also add supplementary materials in appendix, which helps to understand how our method works. We are now doing more experiments with a new version of classification methods as I mentioned. We expect that we can attach more results in the camera-ready if we are accepted.
>
> 2. Unsupervised clustering (even when number of classes is not known) is a very well studied problem in machine learning.
> In this work, we want to show the advantage of classification than clustering. The main difference between classification and clustering is that there exists measurable metrics for each class, which enable us to calculate the probability of belonging in each class. On the other hand, the clustering has to calculate the distance between the whole dataset. It causes the very high computational complexity. In this paper, we show a classification method in the manner of unsupervised learning, which has not been well studied. In our method, we propose the Gaussian distributions for each class of which mean is the eigenvalue of C^TC. This will be a metric for classification in the manner of unsupervised learning.
>
> 3. importance of self-expressiveness
> In the revised version, we made an assumption 3.2 that each class manifold lies in the subspace of R^n., and vectors in different class are disjoint. We also showed that the connector C is a convex-compact linear map. It implies that, when we express a data point in a latent space by the linear combination of whole dataset, the coefficients of vectors which is not in the class of the given data points are zero. This can be used as the similarity in spectral clustering method. In this paper, we define the mutual expressiveness as the relationship between basis of H and Z which are in same classes

---

> ### Author Response · Authors · 2018-12-01
> **second comment**
>
> 4. section 3 is confusing
> After the first submission, we find out the result by footprint mask is somewhat unstable, and the theoretical meaning is lack as you mentioned. We find another classification criterion which shares the same theoretical background.
>
> we want to find a disentangled property among each class. We can interpret each C_i for class i into a singular value decomposition. In the experiment, we observed that the changes of magnitude and angle by the linear map C are different with the class factor in appendix. This implies C_i has different distinct singular values and semi axis. Then, each block component C_i in C has different eigenvalues. Therefore, we can assign eigenvalues into block sub-matrices. These eigenvalues will be the metric for classification.
>
> The better aspect of classification than clustering is that the classification provides the score for each class that measures how much likely a given data point belongs to the class. Our method uses the eigenvalue as the classification metric. We define the Gaussian distributions of which mean is the eigenvalue of C^TC for each class. We find eigenvalues as much as the estimated number of classes by our algorithm. We calculate the probability for each class for the change of magnitude of a given data point. Then, we can find which class is most probable.
>
>
> 5. Why can we assume propositions 3.1 and 3.2?
> We first notice that the proposition 3.1 and proposition 3.2 are changed to theorem 3.1 and theorem 3.2.
>
> You mentioned that the proposition 3.1 and 3.2 are not well explained. In the revised version, we tried to show those propositions work, and prove the existence of a linear map C with more realistic assumptions than the union of linear subspaces which is used in the sparse subspace clustering.
>
> We revised the part of explaining the existence of an (approximate) linear map by introducing the lemma 3.1. The reason we called C as an approximate linear map is that we do not solve the exact dimension of the subspace which class manifolds lie on. we empirically observe that the dimension of the subspace that the subset (class manifold) lies on is approximately (the dimension of latent space) / (the number of classes).
>
> We changed the hypothesis to the assumption 3.1, and introduce the assumption 3.2. Assumption 3.1. tells us that we can get a tuple (x, h, z), which shares the same factors such as class, brightness, sheerness, sharpness, by using D, C, G after training networks with the cycle consistency loss. That is to say, an data point x encodes into two latent vectors in H, Z respectively, without losing its information. This is because the cycle consistency loss makes the networks encode the right permutation. Therefore, we can track the transformations of the class manifold through the different domains X, H, Z by using D, C, G.
>
> To show the existence of a convex-compact linear map between two class manifolds in H and in Z, we introduce Assumption 3.2. Assumption 3.2 is more realistic and tight than the assumption that the data space is the union of linear subspaces. Each class manifold in the high abstraction space (H and Z) has convexity and compactness as discussed in Bengio et al. 2013. In addition, we showed that class manifolds are disjoint, so we cannot generate other class samples by using a few samples in a class. It implies that the subspaces that the two different class manifolds lie on are orthogonal and sums of the dimension of subspaces H_i and Z_i are dim(H) and dim(Z) respectively. Therefore, we can conclude that H and Z are the direct sums of H_i and Z_i.
>
> By introducing the lemma 3.1. we know that C_i is convex and compact linear map. Then, we can prove the existence of a convex-compact linear map. This linear map guarantees at least between S_H_i and S_Z_i. Even if C_i does not hold compactness and convexity outside the the class subset, it does not matter. Because the outside of class manifolds is nothing.
>
> Now, we can find a linear map from H to Z, which satisfies in the union of class manifolds, by using Theorem 3.2.

---

### Meta-Review · Area_Chair1 · 2018-12-15
**Not ready for presentation at ICLR**

**Confidence:** 5
**Recommendation:** Reject

**Metareview:**

Following the unanimous vote of the submitted reviews, this paper is not ready for publication at ICLR. Among other concerns raised, the experiments need significant work, and the exposition needs clarification.